



# A Reel-Down Instrument System for Profile Measurements of Water Vapor, Temperature, Clouds and Aerosol Beneath Constant Altitude Scientific Balloons

Lars E. Kalnajs[1], Sean M. Davis[2], J. Douglas Goetz[1], Terry Deshler[1], Sergey Khaykin[3], Alex St. Clair[1], Albert Hertzog[4], Jerome Bordereau[5], Alexey Lykov[6]

[1]Laboratory for Atmospheric and Space Physics, University of Colorado at Boulder, Boulder, Colorado, 80303, USA
[2]NOAA Chemical Sciences Laboratory, Boulder, Colorado, 80305, USA
[3]LATMOS/IPSL, UVSQ, Sorbonne Université, CNRS, Guyancourt, 78280 France
[4]Laboratoire de météorologie dynamique, Sorbonne Université, Palaiseau, 91128, France
[5]Laboratoire de météorologie dynamique, CNRS, Palaiseau, 91128, France
[6]Central Aerological Observatory of Roshydromet, Dolgoprudny, 141700 Moscow Region, Russia

*Correspondence to*: Lars Kalnajs (kalnajs@colorado.edu)

**Abstract.** The Tropical Tropopause Layer (14 - 18.5 km) is the gateway for most air entering the stratosphere, and therefore processes within this layer have an outsized influence in determining global stratospheric ozone and water vapor concentrations. Despite the importance of this layer there are few in situ measurements with the necessary detail to resolve the fine scale processes within this region. Here, we introduce a novel platform for high resolution in situ profiling that

lowers and retracts a suspended instrument package beneath drifting long duration balloons in the tropics. During a 100-day circumtropical flight, the instrument collected over 100 two-kilometer profiles of temperature, water vapor and aerosol at one-meter resolution, yielding unprecedented geographic sampling and vertical resolution. The instrument system integrates proven sensors for water vapor, temperature, pressure and cloud and aerosol particles with an innovative mechanical reeling and control system. A technical evaluation of the system performance demonstrated the feasibility of this

new measurement platform for future missions with minor modifications. Six instruments planned for two upcoming field campaigns are expected to provide over 4000 profiles through the TTL, quadrupling the number of high-resolution aircraft and balloon profiles collected to date. These and future measurements will provide the necessary resolution to diagnose the importance of competing mechanisms for the transport of water vapor across the TTL.

## 1.    Introduction


Superpressure balloons are emerging as a practical platform for Earth science observations at altitudes from the upper troposphere to mid- stratosphere, carrying payloads ranging from a few up to 1000's of kilograms with durations ranging



from weeks to close to a year. The balloons, drifting on an isopycnic surface, provide measurements along a quasi-Lagrangian trajectory.

The Reeldown Aerosol Cloud Humidity and Temperature Sensor (RACHuTS) is an instrument system developed to perform profile measurements of the atmosphere up to 2 km below a superpressure balloon (Fig 1). The instrument system consists of three primary assemblies, a reeling system that is contained within the primary balloon gondola, a smaller sub-gondola or profiler that is lowered from the primary gondola and a suite of sensors within the profiler that measure position, temperature, pressure, water vapor and aerosol and cloud particles. The RACHuTS instrument was specifically designed for the Stratéole 2 field experiment – a series of long duration ballooning campaigns to study the Tropical Tropopause Layer (TTL) using a constellation of superpressure balloons circling the Earth at the equator (Haase et al., 2018). The Stratéole 2 experiment is comprised of three field missions: a recently completed engineering test campaign (November 2019 – March 2020) with 8 balloons, the first science campaign in 2021 with 20 balloons and a second science campaign in 2024 in the opposite phase of the Quasi Biennial Oscillation (QBO), also with 20 balloons. A total of seven RACHuTS instruments will be deployed during the experiment; one was deployed during the engineering test campaign, and three will be deployed on each of the upcoming science campaigns. This work describes the RACHuTS instrument design and operation, the technical performance of the overall instrument system during the Stratéole 2 engineering test campaign, and the scientific performance of the sensors within the profiler.

## 1.1 Background

The Stratéole 2 mission builds on the technical accomplishments of three long duration superpressure balloon campaigns in the Antarctic and tropical lower stratosphere. The Vorcore experiment, in 2005, flew 27 superpressure balloons from McMurdo Station, Antarctica, at altitudes between 17.5 and 19.5 km (Hertzog et al., 2007). The Concordiasi campaign in 2010 flew 19 superpressure balloons from McMurdo Station carrying more extensive instrument payloads than used during Vorcore (Boullot et al., 2016; Hoffmann et al., 2017; Rabier et al., 2012). As part of the preparations for the Concordiasi campaign, 3 engineering test flights were conducted from the Seychelles during the Pre-Concordiasi campaign (Podglajen et al., 2014). Stratospheric winds near the equator are primarily zonal, with the direction determined by the altitude and the phase of the QBO, thus balloons released into the tropical lower stratosphere will tend to circumnavigate the tropics (Baldwin et al., 2001). Pre-Concordiasi validated the concept of flying superpressure balloons in the tropics, each of the three balloons flew for longer than 3 months, with one making a circumnavigation of the globe, while the other two balloons reversed course as the phase of the QBO reversed. These engineering flights served as the pathfinder for a dedicated equatorial balloon experiment, Stratéole 2, with a focus on observing the TTL.

The TTL is a region of particular scientific interest, but also suffers from a scarcity of scientific observations. In contrast to the abrupt transition from the troposphere to stratosphere that is typical of the mid-latitude tropopause, the tropical tropopause layer is a 4 km thick transition region, having properties of both the troposphere and stratosphere (Fueglistaler et al., 2009). The TTL forms a gateway to the stratosphere, as most of the air that enters the global



stratosphere passes through the TTL, thus the TTL strongly impacts the composition of the global stratosphere. The microphysics, thermodynamics and dynamics of the TTL are complex and inhomogeneous, driven by powerful tropical convection that impacts the TTL through detrainment, mixing, and wave disturbances. While of great scientific importance, processes in the TTL are difficult to observe, leading to a scarcity of detailed measurements. The altitude of the TTL, 14 –

18.5 km, is above the flight ceiling for most research aircraft, with only a handful of aircraft experiments focusing on this region (Jensen et al., 2015). The region is a thin, cold and dry layer on top of the thick, warm and wet tropical troposphere, making it difficult to observe from satellites. Many of the processes within the TTL occur at fine vertical scales, much smaller than is resolvable by space borne sensors (Randel and Jensen, 2013). Finally, many of the countries in the equatorial belt are developing nations, with few resources to allocate to making regular balloon borne soundings of the TTL. As a

result, there is only a single tropical site performing routine water vapor soundings from Costa Rica (Selkirk et al., 2010), a few campaign measurements over South America, and no measurements over Equatorial Africa.

The Stratéole 2 experiment is designed to overcome many of these limitations by making high resolution in situ and remote sensing measurements of the TTL from a constellation of superpressure balloons circumnavigating the tropics. The Stratéole 2 balloon platform consists of 11 and 13 m diameter superpressure balloons each carrying two gondolas. The

upper gondola (the 'Euros'), manufactured and operated by the Centre National d'Etudes Spatiales (CNES), carries the flight control and flight safety equipment. For safety reasons the Euros is completely isolated from the lower gondola (the 'Zephyr'), designed and operated by the Laboratoire Atmosphères, Milieux, Observations Spatiales (LATMOS), the Laboratoire de Météorologie Dynamique (LMD), and Division Technique de l'INSU (DT-INSU), which hosts the scientific instruments. Each Zephyr can host 2-3 instruments, with a total instrument mass of 8 kg and a total average power

consumption of 10 W. The Zephyr provides the infrastructure to host the instruments, including batteries, solar energy system, thermal regulation and bi-directional communications through an Iridium satellite link. The Stratéole 2 balloons fly at two density levels. The 11 m balloons fly at a density level of 125 g m$^{-3}$, or approximately 18.5 km, and are targeted at in situ observations at the top edge of the TTL. The 13 m balloons fly at a density level of 100 g m$^{-3}$ or approximately 20.5 km, in the lower stratosphere, and primarily host remote sensing instruments. The balloons are expected to fly for 90 – 120

days and will circumnavigate the globe within a flight domain of 20° S to 15° N.

Given that Stratéole 2 balloons are designed to maintain neutral buoyancy, in situ measurements are limited to a single density level. RACHuTS is designed to overcome this restriction by probing the atmosphere down to 2 km below the flight level of the balloon. Deployment of a reel-down sub-gondola from a balloon in flight is complex both from a technical and regulatory stand point. There are few previous examples of this technology in the literature, and prior work is limited to

systems that were not deployed on an operational basis. Hazen and Anderson (1985) describe a large winch system that was deployed on a zero pressure balloon on at least two occasions, deploying a 62 kg payload 12 km below a drifting zero pressure balloon at an altitude of 39 km over Palestine, Texas. At about the same time, a smaller winch system was developed in Japan to deploy a 22 kg payload 3 km below a zero pressure balloon in flight (Matsuzaka et al., 1985). Miniaturized instruments, advanced manufacturing and materials, and improvements in battery technology have made



smaller, lighter and less expensive reel down systems a reality for use on constellations of small balloons without the prospect of instrument recovery.

## 1.2 Design Criteria

The design of the RACHuTS instrument is a compromise between science requirements, logistical constraints of the Zephyr, regulatory requirements for international air space, and safety requirements from the balloon operator, CNES, to insure the survival and safe operation of the balloon system. The primary scientific requirements are high vertical resolution, sufficient measurement sensitivity for the lower stratosphere and a vertical extent to cover as much of the TTL as possible, including spanning the cold point tropopause. The Zephyr has limited resources for allowable mass, power, data and thermal control. RACHuTS shares a Zephyr with the Laboratory for Atmospheric and Space Physics (LASP) Particle Counter (LPC), leaving RACHuTs a mass budget of 6 kg, daily average power of 5 W, and daily data downlink budget of 1.2 MB.

The International Civil Aviation Organization (ICAO), maintains the Convention on International Civil Aviation which includes internationally adopted Rules of the Air (Annex 2) governing the operation of aircraft (ICAO, 2005). The Rules of the Air govern the operation of 'unmanned free balloons' in international airspace. There are negligible regulations for the operation of an unmanned balloon above an altitude of ~18 km (FL 600), where the main Stratéole 2 balloons fly. However, while the Zephyr will remain above 18 km, the profiler will descend to 16 km, where it is subject to more stringent regulation. There is an exemption to many of these regulations for lightweight balloon payloads such as radiosondes. Thus, by meeting the standards for a lightweight instrument, the profiler is exempt from most restrictions below 18 km. These standards applied to the profiler restrict the mass to less than 3 kg and the fiber, suspending the profiler, to break at an impact force of 230 N.

The deployment of the sub-gondola from the main gondola during flight has consequences for the operation and safety of the overall balloon system that impose further requirements. The two primary concerns are (i) accidental loss of the profiler, and (ii) drag or lift generated by the profiler from wind shear encountered during a profile. Loss of the profiler has the potential to be catastrophic, forcing the balloon to lower ambient pressures and hence to a superpressure environment exceeding the balloon design strength. A similar situation could occur if the deployed profiler encounters significant wind shear. Counter-intuitively, wind shear on a tensioned line will cause lift (Alexander and Stevenson, 2001), reducing the load on the balloon leading to additional balloon overpressure. These concerns led to the following instrument requirements. The profiler must withstand the highest loads expected during flight, typically at launch or flight termination, estimated at 13 times gravity (g), without loss of the profiler, and the mechanical reeling system must be incapable of breaking the line. Finally, the deployment of the profiler initially will be phased and occur in coordination with the flight dynamics managers to allow for the impacts on balloon overpressure to be assessed.



## 130  2.      Instrument Description

RACHuTS consists of three main sub systems: the reeling system, the profiler gondola, and the profiler sensor suite, consisting of a temperature and pressure sensor (TSEN), water vapor sensor (FLASH-B), and cloud and aerosol sensor (ROPC).

## 135  2.1 Reeling System

The reeling system is contained within the Zephyr, and consists of the mechanical assembly to unspool and spool the profiler line, the electro-mechanical interface to the profiler, and the electrical control and communications interface to the Zephyr (Fig 2). The core of the reel system is a 6 cm diameter by 12 cm long spool, which is 3-D printed using Selectively Laser Sintered (SLS) carbon fiber reinforced nylon. The spool is designed to accommodate up to 3 km of line, but was loaded with 2.2 km for the Stratéole 2 engineering campaign. The spool is driven by the primary drivetrain using a brushless DC servo motor and planetary gear box providing 2.4 N m of peak torque at speeds up to 650 rev min$^{-1}$. This translates to a maximum pulling force of 60 N with an empty spool, well below the minimum breaking strain of the line and approximately 3 times higher than nominal gravitational load of 22 N from the 2 kg profiler and line. An electromagnetic friction brake provides 4.8 N m of holding force to the reel. The electromagnetic brake is failsafe, a current must be applied to release the brake and in the event of a loss of power the brake will engage, holding the profiler. A secondary drive is digitally slaved to the primary drive and powers a level wind carriage, which oscillates back and forth across the width of the reel, evenly distributing the line on the reel.

The profiler is suspended by a 280 μm diameter ultra-high molecular weight polyethylene (UHMWPE, trade name Spectra) braided line with a weight of 200 g km$^{-1}$. Multiple sections of line were pull tested to failure with a mean breaking strain of 192 N at 25 °C (std. dev. 5.7 N). One property of particular importance to the measurement environment is the glass transition temperature of UHMWPE and the temperature dependent breaking strain, as the line will be passing through the cold point at temperatures as low as -90°C. The glass transition temperature -150°C is well below the lowest temperatures expected (Sobieraj and Rimnac, 2009), and pull testing was repeated in a thermal chamber, with a maximum yield strength of 204 N reached at -80 °C. This is below the ICAO mandated 230 N breaking strain, but a factor of 10 above the nominal load from the profiler. To meet the flight safety requirement of withstanding a 13 g acceleration (a 260 N force on the line) the first 5 meters of the line were reinforced with a supplemental 1.2 mm diameter UHMWPE braid with a tensile strength in excess of 1 kN, jacketing the primary line. The braided line enters the bottom of the Zephyr through the profiler docking connector. From there, the line passes over a large diameter spring loaded pulley. This kinematic pulley serves two purposes; it provides a mechanism to dissipate shock, as the UWMWPE has a high elastic modulus (low stretch), and it provides over and under load detection. The pulley is spring loaded in two directions, under a high load situation the pulley moves toward the exit orifice, actuating an overload microswitch. If the system becomes unloaded, risking tangling



in the line, the pulley moves away from the exit orifice tripping a second underload microswitch. The line and motor systems were subject to extensive lifetime testing in the laboratory. Using two RACHuTS reel systems on opposite ends of 2 km of line, over 600 profiles were simulated by deploying the line from one reel, while reeling it onto the other reel, then reversing the process.

On the exterior of the bottom of the Zephyr is the profiler docking connector. This unit provides both a mechanical guide to steer the profiler into alignment with the Zephyr as well as electrical connection to the profiler. Four gold rings on the bottom of the Zephyr contact four concentric rings of spring-loaded contact pins on the top of the profiler, providing power and communications with the profiler while docked.

The reel system is controlled by an ARM Cortex microcontroller on the purpose-built Motion Control Board (MCB). The MCB enforces programmable limits on speeds, torques, currents and voltages to the motors while collecting performance data for each profile. The interface between the MCB, the profiler and the Zephyr is provided by a second module, the Profiler Interface Board (PIB), also purpose built around an ARM Cortex microcontroller. The PIB initiates a profile by sending commands to the docked profiler to begin making measurements, then commanding the MCB to begin reeling operations. At the completion of a profile the PIB offloads the stored science data from the profiler, collects the reel performance data from the MCB and then sends the compressed data on to the Zephyr for transmission to the ground station.

## 2.2 Profiler

The profiler, lowered from the Zephyr, obtains power and communications from the Zephyr (through the PIB and docking connector) while docked. A rendering of the profiler comprised of FLASH B, TSEN electronics, ROPC, and rechargeable batteries is shown in Fig 3. The profiler is housed in a 25 cm diameter by 28 cm tall expanded poly-styrene (EPS) cylindrical thermal housing with an overall weight of 1.8 kg. As the profiler is outside the Zephyr, thermal management is controlled internally.

Control and data storage for the profiler is provided by an ARM Cortex microcontroller, which interfaces with each of the sensors via a serial connection, and with the PIB when docked. The control board also includes a temperature measurement system for temperature control and engineering data, two temperature control loops, a GPS receiver, and a flash memory card for data storage. Power for the profiler is provided by a lithium-ion battery pack, with sufficient capacity for five profiles and 24 hours of thermal management. The battery pack is charged from the Zephyr while docked. Lithium batteries can be damaged through lithium plating when charged at temperature below 0 °C (Petzl et al., 2015), thus the battery pack is temperature controlled at 0 °C, while the rest of the profiler is controlled to -20 °C.

### 2.2.1 TSEN Temperature and Pressure Sensor

Temperature and pressure measurements on the profiler are made using the Thermodynamical SENsor (TSEN) instrument, developed and supplied by LMD in Paris, France (Hertzog et al., 2007). The instrument measures ambient air temperature outside the profiler gondola with a small diameter silver coated thermistor suspended on a wire 5 meters below the profiler.

Pressure is measured inside the profiler using a board mounted silicon pressure sensor, with an integrated analog to digital converter. The temperature of the pressure sensor is measured internally to provide a temperature compensated pressure measurement. The TSEN electronics are housed in the temperature-controlled environment inside the profiler and report pressure and temperature at 1 Hz both during the profiles and at flight level.

TSEN has a long flight heritage on superpressure stratospheric balloons and is well characterized in the lower most stratosphere. Prior generations of the TSEN instrument were deployed on multiple superpressure balloons in a similar configuration during the Vorcore campaign in Antarctica in 2005 (Hertzog et al., 2007), on the Concordiasi Campaign in Antarctica in 2010 (Ward et al., 2014; Hoffmann et al., 2017) and in the tropics during pre-Concordiasi also in 2010. The version of TSEN used in the profiler is slightly evolved from the Concordiasi instrument with an improved thermistor with a reduced size to decrease solar bias. Furthermore, the TSEN in the profiler uses the much smaller and lighter MS5803 pressure sensor instead of the larger quartz crystal barometers used in the standard TSEN.

The TSEN temperature sensor accuracy and precision, determined during prior flight campaigns, are 0.1 K at night and 0.25 K during the day when corrected for solar radiation (Hertzog et al., 2004). For the RACHuTS profiles a precision of 0.1 K can be assumed as the profiles are collected at night, and the sensor is ventilated by the vertical motion of the profiler and any horizontal wind shear. The pressure sensor is specified by the manufacturer to have an absolute accuracy of ± 2 hPa, and a precision of 0.01 hPa over the temperature range inside the profiler. A higher absolute accuracy can be obtained by performing cross calibration with the TSEN in the Euros gondola while to profiler is docked.

### 2.2.2  FLASH-B Water Vapor Sensor

FLASH-B is a compact and lightweight version of the FLASH (Fluorescence Lyman-Alpha Stratospheric Hygrometer) instrument developed at the Central Aerological Observatory in Russia specifically for balloon-borne measurements of water vapor in the upper troposphere and lower stratosphere (Yushkov et al., 1998). The instrument senses water vapor by measuring the intensity of fluorescence from OH radicals that have been photodissasociated from water molecules exposed to Lyman-alpha radiation (121.6 nm). The Lyman-alpha is produced by an on-board hydrogen lamp, whereas the fluorescence signal at 308-316 nm is measured using a photomultiplier tube in photon counting mode. The intensity of the fluorescence is directly proportional to the water vapor mixing ratio at stratospheric conditions (Yushkov et al., 1998). FLASH-B uses a coaxial open path optical layout, in which the measurement volume is located outside the instruments, 2-3 cm away from the lens of the instrument. To reduce the background light and to avoid saturation of the photomultiplier tube, FLASH-B can only be operated at night, thus RACHuTS profiles are only collected with a solar zenith angle (SZA) greater than 95°. FLASH-B is installed within the profiler housing with the lens facing downward. Since the lens assembly, exposed to ambient temperature at one end, provides a thermally conductive path for heat loss, FLASH-B is thermally isolated from the other profiler components by an 8mm thick aerogel blanket, and provides its own thermal management, regulating to -20 °C during operation.



FLASH-B has a significant flight heritage on sounding balloons in the tropics (Khaykin et al., 2009) and in the Arctic (Khaykin et al., 2013), as well as in a previous long-duration balloon experiment (Lykov et al., 2014). The performance of FLASH-B has been extensively documented both in laboratory intercomparisons (Fahey et al., 2014), and

through collocated balloon soundings with frost-point, tunable diode laser, and Lyman-alpha hygrometers (Ghysels et al., 2016; Khaykin et al., 2013; Vömel et al., 2007), yielding mean relative deviations less than 2.4% in the lower stratosphere. Prior to delivery, the FLASH-B instrument is calibrated in a stratospheric simulation chamber at constant pressure (50 hPa) and temperature (-40 °C) over a wide range of mixing ratios (1 – 100 ppmv) against the reference dew point hygrometer MBW 373L. The detection limit for a 4-second integration time is of the order of 0.1 ppmv, while the accuracy is limited by

the calibration error amounting to 4%. The typical precision in the stratosphere is 5 - 6%, whereas the total uncertainty is less than 10% throughout the stratosphere. During profiles, the primary measurement data (i.e., the fluorescence signal) is reported and stored at 1 Hz, whereas housekeeping and diagnostic data is reported at 0.1 Hz.

### 2.2.3    RACHuTS Optical Particle Counter

The RACHuTS Optical Particle Counter (ROPC) is a small 8 channel OPC optical head based on the MetOne 9722-1 optical

head.   The ROPC is a closed path instrument that uses a constant volume rotary vane pump to draw 3.5 L min$^{-1}$ of air in through a 3 mm diameter inlet tube and through a laser beam.  Particles in the air stream scatter light from the 670 nm laser diode. The side-scattered light is collected over a cone from 60-120˚ with an elliptical mirror which focuses the scattered light pulses on a photodiode. The electrical signal from the photodiode is digitized and the intensity of each pulse is divided and accumulated into 8 bins by a pulse height analyzer, representing 8 size bins approximately logarithmically spaced from

0.3 – 10 µm diameter.   The rotary vane pump is a constant volume pump and thus pulse frequency is easily converted to aerosol concentration.   After passing through the pump, the air passes through a 0.01 µm filter to remove particles generated by the graphite vanes in the pump, before being exhausted to the atmosphere.  The first size bin, 0.3 µm, is recorded at 1 Hz while the larger channels have an 8 s integration time.

The ROPC has a less extensive stratospheric heritage than the either TSEN or FLASH-B.  A version of the MetOne

9722 optical head is used in the Laboratory for Atmospheric and Space Physics (LASP) Stratospheric Total Aerosol Counter (STAC - a balloon borne Condensation Particle Counter) and has been flown on sounding balloons numerous times over the past three years.   Total aerosol concentration profiles from the STAC have agreed with similar profiles from the University of Wyoming Condensation Nuclei (CN) counter (Rosen and Hofmann, 1977; Campbell and Deshler, 2014) within geophysical variations of CN profiles.  An inflight comparison was also made between the ROPC and the Wyoming laser

particle counter (WLPC) (Ward et al., 2014) in 2018 on a sounding balloon. Below 20 km the instruments agreed to within 20%; however, the ROPC detection efficiency decreased relative to the LPC above 20 km.   Unlike the STAC or WLPC, the ROPC does not use a filtered air sheath flow to constrain the sample air jet, and it is possible that at pressures below 50 hPa the air jet through the laser beam begins to diverge, lowering the counting efficiency.  While this effect is unlikely to impact the RACHuTS measurements that are all below 20km, the ROPC will require further validation.  The ROPC was primarily



included in the payload to detect super-micron cirrus cloud ice particles, while the measurement of the number density of ice particles needs further validation during flight, there is high confidence in the ability to detect cloud particle (d > 1μm) at concentrations above 0.1 L⁻¹.

## 3.        Stratéole 2 Engineering Flight

The first field activity of the Stratéole 2 experiment took place in October to December 2019 at the Seychelles International
Airport on the island of Mahé in the Seychelles (4.67 °S, 55.52 °E). This campaign was an engineering test and proof of concept for the two primary science campaigns scheduled to occur in 2021 and 2024.    During the campaign, one of each balloon configuration was flown, with some additional tests for a total of 8 balloon flights.    RACHuTS was hosted on the 'TTL3' configuration of the Zephyr, sharing the gondola with an LPC and using an 11 m diameter superpressure balloon. The TTL3 flight was launched at 19:20 UT on November 18 and reached a stable float altitude of 18.8 km the following day.
The flight lasted 101 days and was terminated on 28 February 2020 off the East coast of Ecuador (3.33 °S, 81.42 °W) after completing 1.5 circumnavigations of the Earth (Fig 4).

       Due to calm winds at launch, necessitating the rapid movement of the gondola, the release of the Zephyr was more dynamic than planned, leading to minor damage to the RACHuTS docking connector. The profiler oscillated below the gondola as the launch team had to both hold the gondola high and pivot the balloon, ascending over the release point,
towards them.    As a result, multiple manual Tele-Commands (TC) and re-docking procedures were required to re-establish communications with the profiler unit. Communications were re-established on November 21, and instrument commissioning began with phased manual profiles.    No negative consequences were observed on the dynamics of the balloon system, and the first full 2 km long science profile was performed on November 23.

       The instrument operational plan specified switching to an automatic profiling mode soon after commissioning.    In
automatic mode, RACHuTS would autonomously begin profiling when the SZA surpassed 95°.    To begin a profile, the profiler requests permission from the Zephyr to commence profiling. Once granted, the profiler enters warm up mode, nominally 15 minutes with all the instruments in the profiler running. Once warm up is complete, the reel deploys 7500 revolutions, lowering the profiler 2 km below the Zephyr at a vertical speed of -1 m s⁻¹.    At the completion of the reel out, the instrument enters a 15-minute dwell period at the bottom of the profile, and the reel performance data from the MCB is
telemetered to the ground. The profiler is then retracted 1.95 km at 1 m s⁻¹. The final 50 m approach to the gondola is performed at 0.1 m s⁻¹, until the torque feedback indicates the profiler is docked. At this point the profiler offloads the approximately 200 kB of profiler data to the Zephyr, which sends it to the ground along with the reel performance data from the ascent.    Once data offload is complete, power to the profiler is turned off and the battery is recharged for the next profile, or for entering a daytime hibernate/charge mode if the SZA < 115°.

290        The mission specification for the RACHuTS flights is 3-4 cycles per night, yielding 6-8 profiles through the TTL as data are collected both on descent and ascent.    Unfortunately, due to the intermittent communications through the docking



connector and a compounding software error, all the cycles during the engineering flight were commanded manually using TCs from the ground. Fifty-five full cycles were completed over the course of the flight, yielding 110 profiles through the TTL and meeting the goals of the engineering campaign by demonstrating the instruments functionality and assessing the performance of the measurements. Had the profiler operated nominally in automatic collection mode there would have been 300-400 full cycles.

**3.1 Reel System Performance**

As an entirely new and complex system, validating the performance of the RACHuTS reeling system was one of the primary goals of the engineering test flight.   Motor performance data for all 110 2-km motions were consistent with pre-flight engineering analysis and laboratory tests.   During the ascent portion of each cycle, reel power consumption averaged 39 W with a motor current of 2.6 A at a nominal voltage of 15 V.  During the descent phase of each cycle the reel motor acts as a retarder, counteracting the gravitational acceleration on the profiler and generating net counter-electromotive force. This generated power was shunted to a large power resistor on the electronics box housing to be dissipated as heat and to limit the increase in motor voltage.   A possible concern identified before the flight was the ability to dissipate this heat in the low pressure stratospheric environment, however the reverse current flow in flight was lower than expected (~ 0.1A) and did not negatively impact ambient temperature within the gondola.

The motor diagnostics, including motor current and temperature, were used as indicators of reel system degradation over the course of the engineering flight. Deterioration of the motors, bearings, or pulley train would appear as an increase in the torque, and therefore current, in the main motor and level wind motors. These diagnostics showed no significant change in reel system performance while in flight, and there was no evidence that the reel system would degrade under the planned sampling regime of 4 profiles per night over 90 days.

Motor torque can also provide insight into lift generated by the profiler line under high wind shear conditions, where reel motor torque below nominal would correspond to profiles exhibiting decreased gravitational loads. During the engineering flight, we did not observe decreased torque during any of the profiles, and while we do not have a direct measurement of windspeed from the profiler, the horizontal offset of the profiler relative to the Zephyr provides a proxy for the integrated wind shear over the length of the profile.   Comparing motor torques for profiles with horizontal displacements < 200 m with profiles with horizontal displacements > 1000 m indicates that the impact on the reel system is not statistically significant with a 1-σ variability in motor torque of 4.7% (Fig 5). It should also be noted that the CNES flight team did not observe significant changes in the TTL3 flight level during an active RACHuTS profile, further demonstrating that wind shear induced lift is not a concern for future Stratéole 2 missions.

Two significant anomalies were observed, the previously mentioned damage to the docking connector and anomalous sensor readings causing the reel to default to a safe mode.  The damage to the docking connector led to intermittent failures in the docking procedure after profiles, which necessitated manual intervention from the ground station to rectify and often required several days for recovery.   Intermittent, but persistent, erratic sensor readings in the reel



control system caused the reel to enter a safe mode, again requiring intervention from the ground to rectify. The operational impact of these two anomalies was the inability to use automatic mode, and a significant reduction (~75%) in the total number of profiles collected relative to the operational plan.

The issues with the damage and intermittent communication through the profiler docking connector and erratic sensor readings are being addressed for future deployments. The connector has been redesigned to use more robust spring

pins, and an additional fixture has been added to the spring pins to provide enhanced mechanical support. A Long Range (LoRa) radio link will be added between the profiler and the PIB to provide continuous communications with the profiler, independent of the mechanical docking. In addition to redundant docked communications, the LoRa link also provides continuous communications during a profile, decreasing the latency of downlinking profile data. The temperature sensing system in the reel system and profiler have been revised to use digital temperature sensors to reduce erroneous readings, and

the firmware has been modified to prevent erroneous readings from triggering safety interlocks.

## 3.2 TSEN Performance

The performance of the sensors in the profiling unit is assessed based on two criteria, the technical operation of the sensor and, where possible, a comparison of the scientific measurement with independent measurements. The profiler TSEN operated nominally throughout the flight, both during profiles and while docked. TSEN collected 3.1 million measurements

of pressure and temperature while docked, and 300,000 measurements during profiles. No anomalies were identified in the instrument operation, and missing data from the flight is due to communications issues with the profiler rather than to the TSEN instrument.

The measurement performance of TSEN is well characterized from prior long duration flights. Two comparisons have been identified for the RACHuTS TSEN: comparison with the TSEN located on the Euros gondola for flight level data,

and comparisons with nearby radio soundings for profile data. Both comparisons have limitations. The positioning of the TSEN temperature sensor on the Euros gondola was dictated by logistical constraints and was not optimal for temperature measurements. The sensor was approximately 1 m above the gondola. This resulted in significant contamination due to solar heating of the Euros gondola re-radiating to the temperature sensor. For all docked data points the RACHuTS TSEN had a mean offset of -0.69 K relative to the Euros TSEN. When considering only the night time measurements, (SZA >

95°) the relative bias drops to -0.46 K. The pressure sensor used in the RACHuTS TSEN had a 3.84 hPa mean positive bias relative to the high accuracy Quartz barometer on the Euros TSEN after applying the manufacturers supplied calibration to the docked TSEN data. The RACHuTS TSEN pressure sensor was operated outside the specified ambient temperature and pressure range to obtain optimum accuracy, leading to a decrease in absolute accuracy. The offset of the RACHuTS TSEN pressure measurement is a strong function of the temperature of the pressure sensor module. After applying an empirical

linear correction to the pressure sensor as a function of the temperature of the sensing element and accounting for the 7 m vertical separation between the sensors, the mean offset between the two pressure sensors was reduced to 0.08 hPa with a 1-σ variance of 0.55 hPa. This correction was then applied to all the RACHuTS TSEN pressure measurements.



Comparisons between RACHuTS profiles and radiosondes were rare due to the scarcity of radiosonde sites within the tropics. Fortuitously, two RACHuTS profiles over Brazil were within 300 km of three operational radiosonde sounding sites (Fig 6). The profile collected on 3 Jan 2020 (Fig 6b) was within 300 km of two radiosonde soundings, however there was significant horizontal and temporal variability in the TTL as is evident in the difference between the RACHuTS descent and ascent profiles, which were spaced 76 km apart, and subsequent radiosonde soundings. Yet the RACHuTS profile is bracketed by the radiosonde profiles. A comparison from the day prior (Fig 6b), shows significantly less spatial and temporal variability, with both the descent and ascent profiles as well as 3 soundings all agreeing within 3 K. The RACHuTS profile shows significantly more detail than is available from the sounding, but the position of the CPT agrees within 50 m, as well as a second inflection at 18.6 km.

### 3.3 FLASH-B Performance

The technical performance of FLASH-B throughout the flight was highly stable and within specification. The instrument was located outside the primary thermal housing and performed its own thermal control, with the lamp temperature reaching the nominal operation range within the 15 minute pre-flight warmup period. The voltage and current supply to the hydrogen lamp were stable throughout the flight, indicative of highly stable Lyman-alpha emission which is critical to the measurement. As was expected, the initial measurements from FLASH-B showed anomalously high water vapor in the vicinity of the balloon (within several hundred meters). This water vapor contamination is due to water vapor outgassing from the balloon and gondola surfaces and is unsurprising given the low ventilation environment (Ghysels et al., 2016; Zander, 1966). The TTL3 balloon was launched into heavy cloud cover and it is suspected that the balloon was coated in ice during ascent. A coincident launch of the NOAA frostpoint hygrometer (FPH) measured a total column water vapor of 52 mm and measured a deep, saturated layer in the upper troposphere, which likely exacerbated the expected water vapor contamination. However, the water vapor contamination was largely limited to measurements early in the flight and to pressures below 67 hPa. During deployments, the profiler is ventilated by its own vertical motion as well as by wind shear encountered below the balloon. The measurements during the ascent of the profiler are more affected by outgassing as the downward looking optics of FLASH-B senses the air coming down from the instrument surfaces.

The scientific performance of FLASH-B can be assessed through comparison with co-located MLS measurements. Figure 7 shows a comparison of the relative frequency of FLASH-B water mixing ratio measurements during the descent phase of the profile to the mean water vapor mixing ratio derived from Microwave Limb Sounder (MLS) profiles within 300 km and 24 hours of the RACHuTS profiles. Greater than 90% of the FLASH-B measurements lie within the 1σ variability of the MLS profiles, in spite of the limited vertical resolution of the MLS water vapor retrievals. Evidence of water vapor contamination from the balloon and gondola is present at pressures less than 67 hPa, potential temperatures greater than 425 K.



### 3.4    ROPC Performance

ROPC operated nominally throughout all the profiles, collecting approximately 300,000 measurements during the warm up, profile and dwell periods.   The only technical issue related to ROPC was an occasional erroneous temperature reading from the thermistor on the rotary vane pump that required correction in post processing.

Independent in-flight validation of the ROPC is difficult to achieve as satellites cannot retrieve aerosol size distributions and balloon borne aerosol soundings are not routinely performed in the tropics.   However, the LPC instrument,

flow rate ~20 L m$^{-1}$, was also hosted on the same gondola as RACHuTS, providing an in situ comparison for the ROPC while it is docked at flight level in the warm up preceding each profile.  Six measurement periods were identified when both LPC and ROPC were operating nominally and made measurements within 15 minutes of each other.  The size distributions from LPC were re-binned to match the channels from ROPC and the concentrations for each ROPC channels were compared.   The total ROPC concentration (all particles > 0.3 μm) was on average 0.54 of the concentration of particles >

0.3 μm from LPC.    The 1.0 μm channel was the largest ROPC channel to routinely detect particles at float altitude and was 0.25 the concentration of the LPC 1.0 μm channel.   If the LPC size distributions are re-binned at sizes ~20% larger than the ROPC nominal channels, ROPC is 0.85 times LPC for the 0.3 μm channel and 1.05 times for the 1.0 μm channel, well within the ROPC uncertainty.  This suggests that ROPC is slightly under sizing particles relative to the LPC.

### 4.    Example of the scientific measurements

As mentioned above, in spite of the engineering difficulties encountered the profiler completed 55 descent/ascent profiles beneath the Zephyr, which maintained an altitude near 19 km throughout the flight. The location of these profiles, which span the Central Pacific to Indian Oceans, are shown in Fig 4. An example profile collected in the Eastern Pacific illustrates several features of scientific interest and relevant to the scientific mission.  Figure 8 displays the temperature, aerosol/cloud, water vapor, and ice supersaturation which were measured on this profile. The RACHuTS profiler descends to ~ 800 m

below the cold point tropopause (CPT), Figure 8a.   The descent and ascent each require 30 minutes and the dwell at the bottom lasted 15 minutes. In that time the profiler moved 37 km to the north east. The temperature of the upper troposphere is very homogeneous throughout this area, and the temperature offset above the tropopause is consistent with the isentropic gradient between the ascent and descent profiles.    Above the CPT, two temperature perturbations are evident in the ascent profile at 18.4 and 18.9 km.   Collocated with the cold phase of these wave driven temperature perturbations there is an

enhancement of large (d > 3 μm) particles (Figure 8b), most likely cirrus cloud ice particles, confirming the wave driven cirrus cloud formation observations obtained during ATTREX (Kim et al., 2016).   Further cirrus cloud ice particles are observed near the CPT on both descent and ascent, suggesting the presence of a sub-visible cirrus cloud near the CPT with significant horizontal extent.    Surprisingly, water vapor mixing ratios are essentially constant across the CPT (Fig 8c),



instead significant super saturations with respect to ice are observed (Fig 8d). A detailed analysis of the scientific results

from the campaign will be presented in forthcoming publications.

## 5.        Summary

The successful deployment of the RACHuTS instrument on Stratéole 2 engineering test campaign in 2019-2020 demonstrates the value of profiling instruments for use on long duration balloons. In spite of the campaign being focused on

engineering data and testing, the RACHuTS instrument produced an unprecedented number (110) of vertical profiles of water vapor, temperature and clouds/aerosol through the TTL. These research quality measurements not only validate the instrument concept, but also directly address the science questions that motivated the design of the instrument.

The technical issues observed in the RACHuTS instrument during the campaign have been addressed with minor design revisions that will be implemented prior to the first Stratéole 2 science campaign in 2021. Three more RACHuTS

instruments are in production for launch in 2021, and further 3 instruments are planned for the second science campaign in 2024.

**Author contribution**

LEK led the instrument design, assembly and test and wrote the manuscript.

AH, JB, provided the TSEN sensor.

AL, SK, provided the FLASH-B sensor.

JDG assisted in instrument design and assembly and led the laboratory testing of the instrument.

SMD, LEK, and TD conceptualized the instrument and analysed instrument data.

DAS developed the instrument firmware and assisted with assembly and test.

LEK, TD, JDG, SMD, SK, DAS, AH & JB participated in the field campaign.

AH is scientific lead of the Stratéole 2 project.

**Competing interests**

The authors declare that they have no conflict of interest.


**Acknowledgments**

The authors gratefully acknowledge the support of the Stratéole 2 science teams, the LATMOS/LMD gondola engineering team, the CNES balloon team, and the Seychelles Meteorological Agency. This work was funded by the NSF under award 1643022, with in kind support provided by CNES and CNRS. Special thanks to Philippe Cocquerez and Stephanie Venel

for making the flight of this unorthodox instrument possible on a CNES balloon.



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

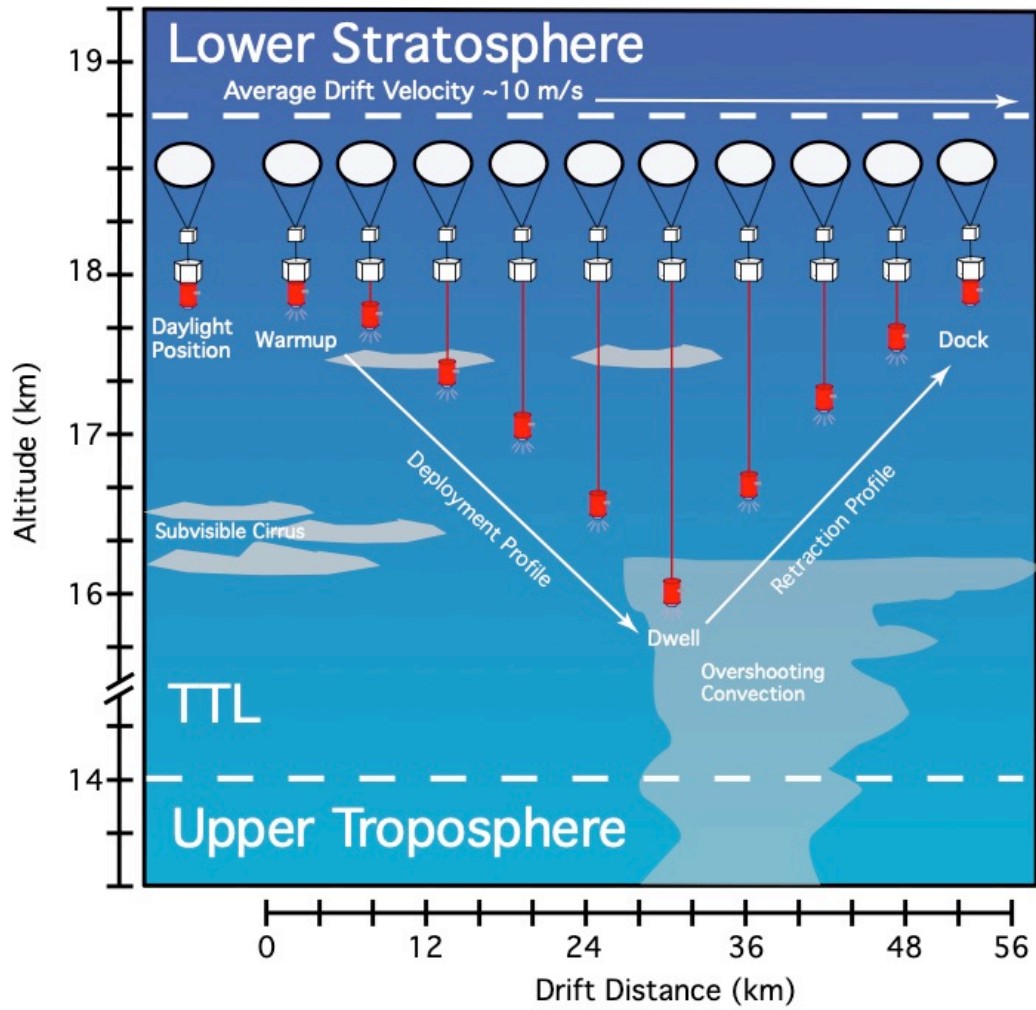

**Figure 1: RACHuTS Operational Concept. The RACHuTS Profiler is deployed from a super pressure balloon at 18.5km, down to an altitude of 16.5km 3-5 times per night making measurements of water vapor, aerosol, cloud particles, temperature and pressure.**



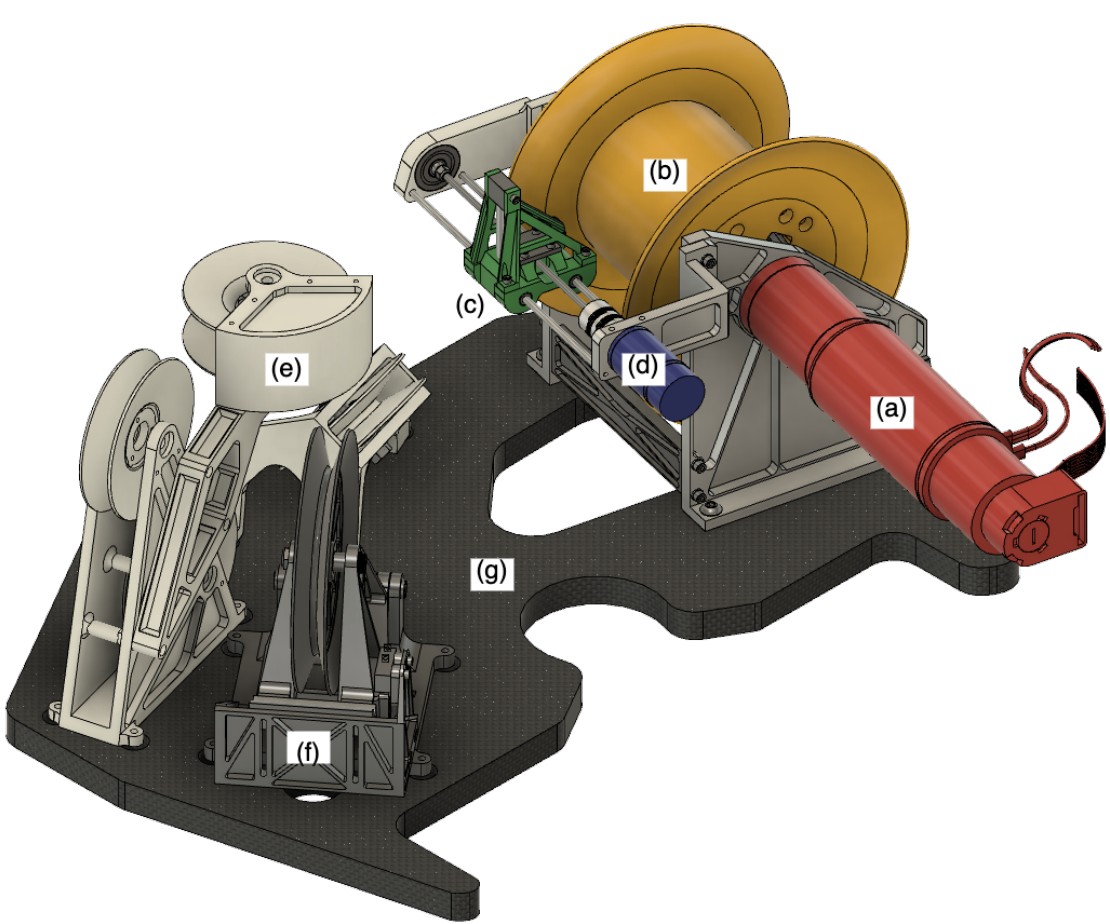

**Figure 2: Rendering of RACHuTS reel system. Major assemblies include the primary drivetrain (a/red), reel (b/yellow), level wind carriage (c/green), level wind motor (d/blue), line redirection pulley train (e/white), kinematic pulley (f/grey) and carbon fiber baseplate (g/black twill), line, electronics box and docking connector are not shown.**






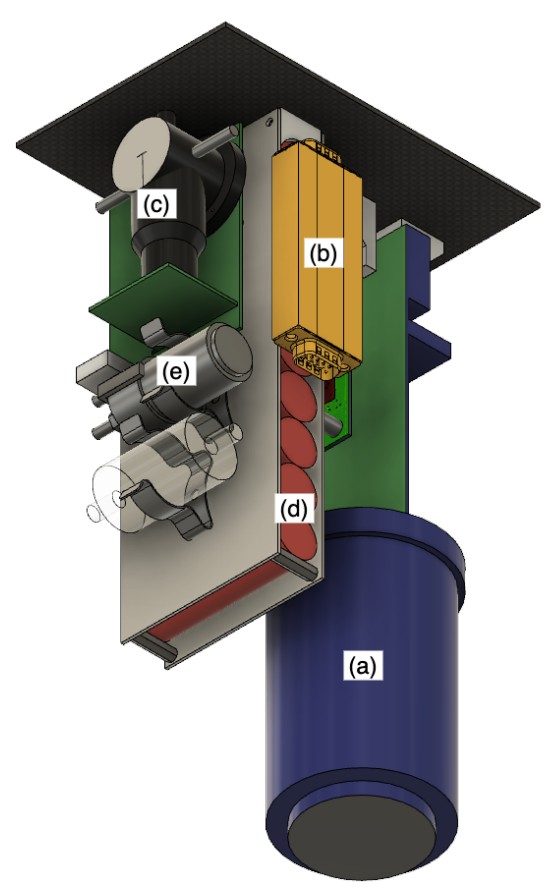

**Figure 3: Rendering of RACHuTS profiler. The FLASH-B water vapor sensor (a/blue), TSEN pressure sensor and electronics (b/yellow) and ROPC aerosol and cloud particle detector (c/black/green) are visible. Other components include the lithium-ion battery pack (d/red), ROPC pump (e/silver). The TSEN temperature sensor and main control board are not visible. The thermal housing and docking connector are omitted for clarity.**






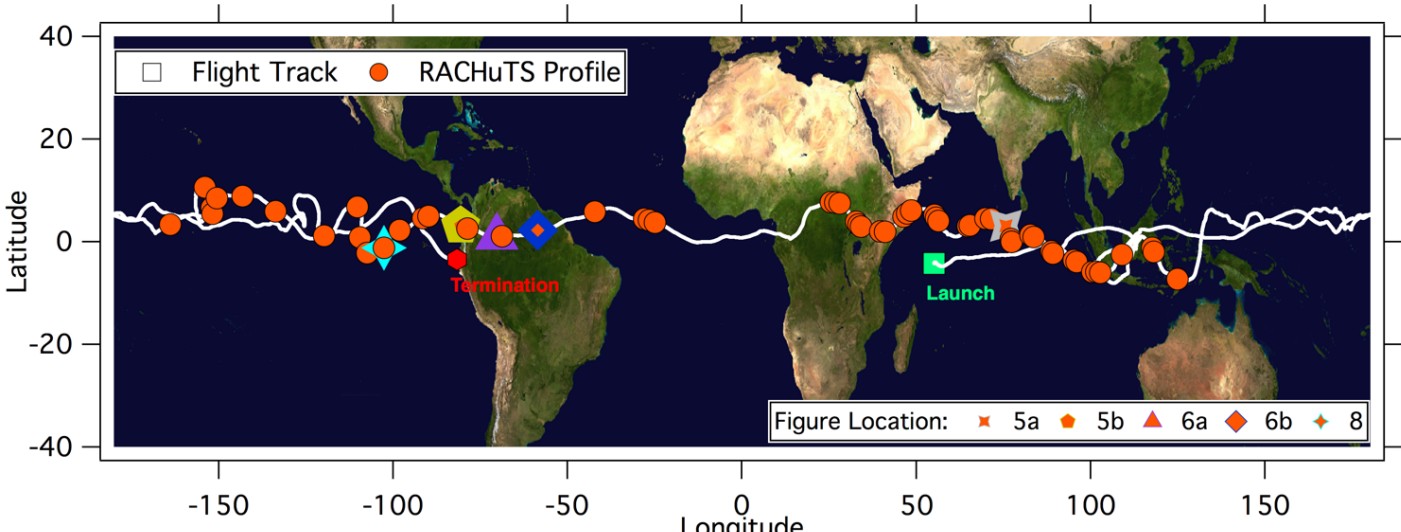

**Figure 4: Flight trajectory of the TTL3 balloon during the Stratéole 2 Engineering test. The trajectory represents 101 days of flight, each point is the position of a 2km RACHuTS profile. Locations of the profiles in figures 5, 6 and 8 are marked. Base map courtesy of NASA Visible Earth.**


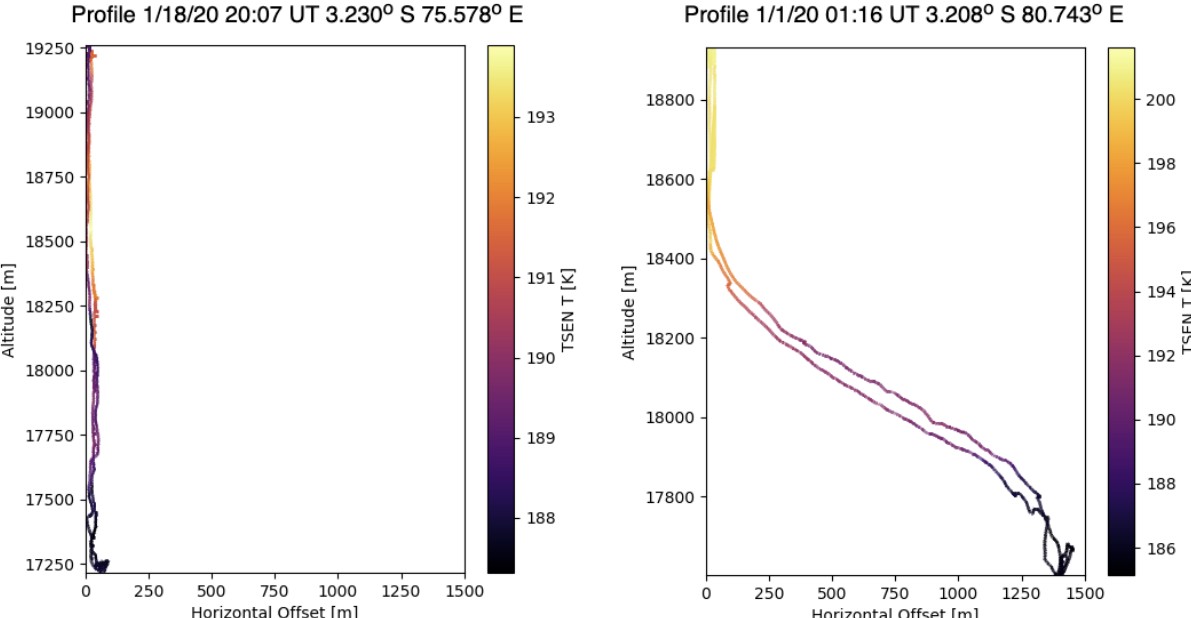

**Figure 5: Horizontal offset of the profiling unit relative to the Zephyr gondola, color coded by ambient temperature. (a) A low**
**wind shear example demonstrating < 100m horizontal displacement and a nearly isothermal profile. (b) High wind shear example**
**with 1400m horizontal displacement and 14K temperature gradient. The difference in motor torque between these profiles is**
**4.7%.**






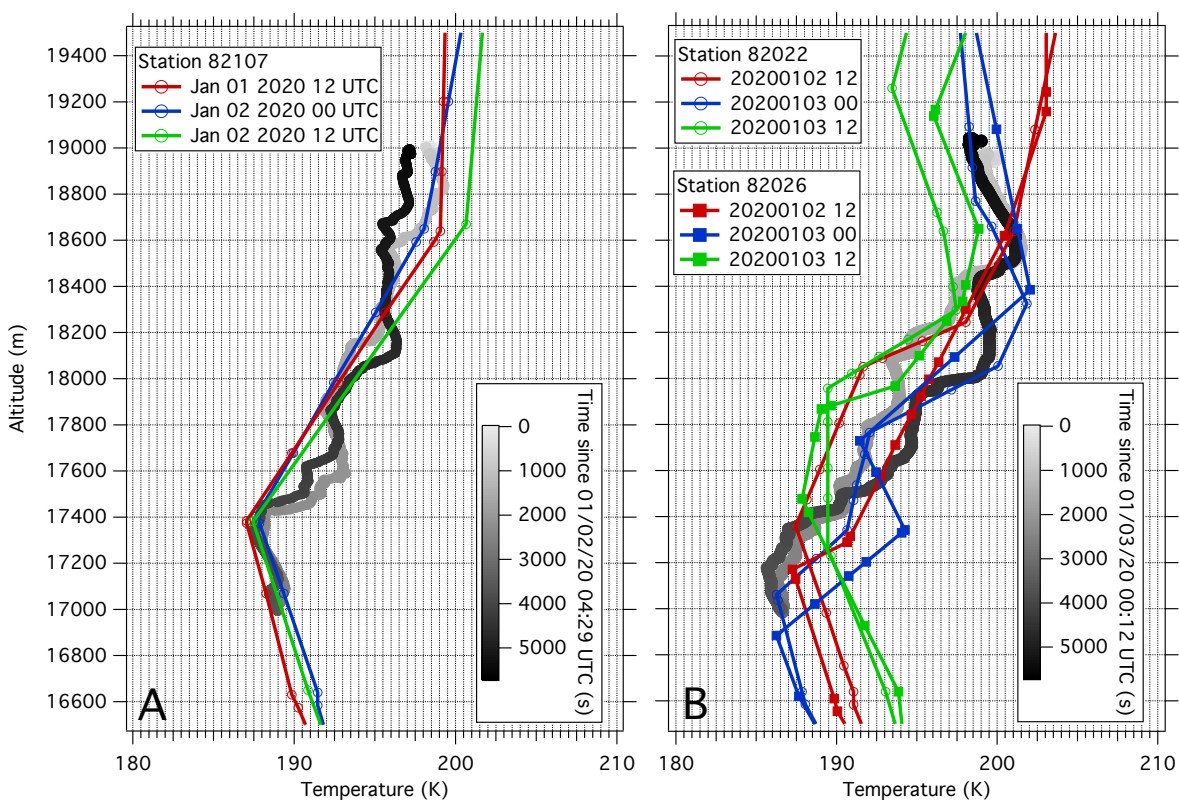

**Figure 6: Comparison between RACHuTS temperature profiles and radiosonde soundings. (a) RACHuTS profile collected at 04:50 on Jan 1 2020 (1.08, -68.75) and three closest in time radiosonde soundings from São Gabriel da Cachoeira, Brazil (-0.12, -66.97, 239 km from profile). (b) RACHuTS profile collected at 00:12 on Jan 3 2020 (2.10, -58.66) and three closest in time radiosonde soundings from Boa Vista, Brazil (2.83N, -60.70, 240km from profile) and Tirios, Brazil (2.22, -55.95, 302km from**
**profile).**



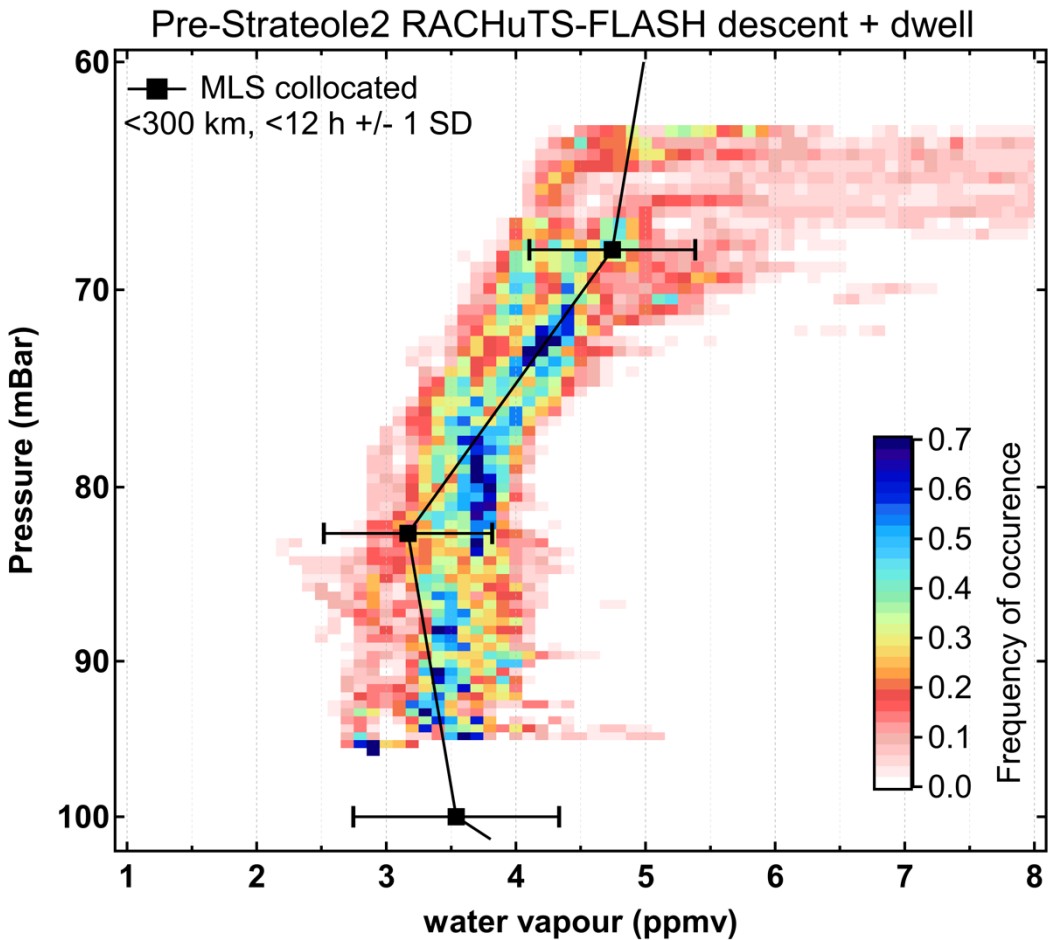

**Figure 7: Comparison of the frequency of FLASH-B water vapor measurements from the 55 descent profiles with the average of closest Microwave Limb Sounder water vapor measurements within 24 hours of each profile.**

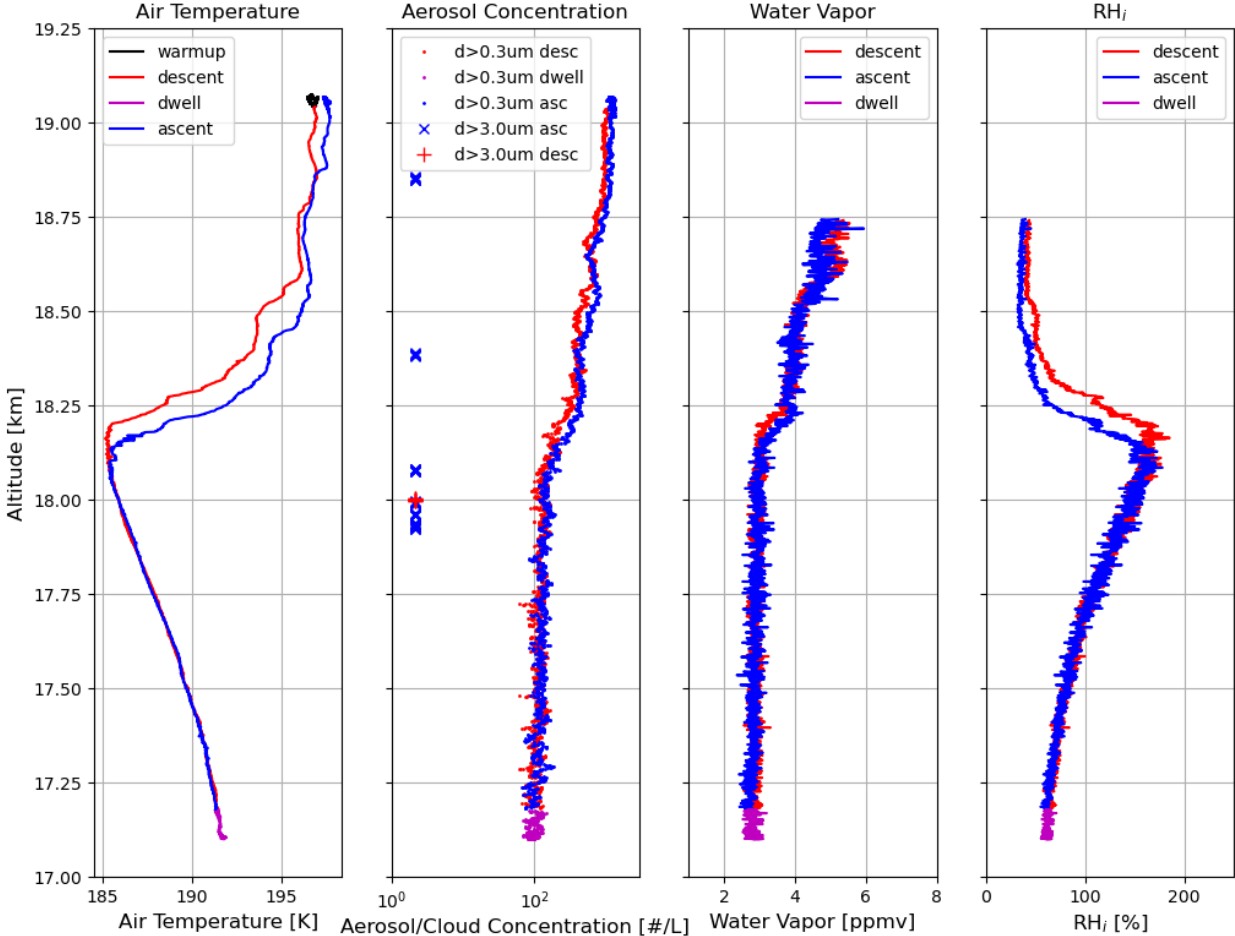


**Figure 8: RACHuTS profile measurement collected in the Eastern Pacific.** All plots are a function of GPS altitude (left to right): (a) TSEN air temperature, note the clear cold point (185K) at 18.2km and wave driven temperature structures near 18.4 and 18.9km on ascent. (b) ROPC Aerosol/Cloud concentration, not the presence of large particles (d > 3.0μm), likely cirrus cloud ice particles, near the cold point (18.2km) and associated with negative temperature anomalies at 18.4 and 18.9km. (c) FLASH-B
water vapor mixing ratio, note the relatively constant mixing ratio across the cold point (18.2km). (d) Calculated relative humidity over ice, near the cold point super saturations exceed 150% suggesting minimal dehydration through cloud formation.