# Peer review of "A Reel-Down Instrument System for Profile Measurements of Water Vapor, Temperature, Clouds and Aerosol Beneath Constant Altitude Scientific Balloons"

_Atmospheric Measurement Techniques, 2020_

## Referee Comment (RC1) · Anonymous Referee #2 · 26 Nov 2020

Review of

**A Reel-Down Instrument System for Profile Measurements of Water Vapor, Temperature, Clouds and Aerosol Beneath Constant Altitude Scientific Balloons**

by Kalnajs et al.

This manuscript presents the successful first deployment of a new, innovative reeling system in combination with proven sensors that measure in situ high-resolution vertical profiles of water vapor, aerosol and cloud particles as well as ambient conditions in the tropical tropopause layer (TTL) from drifting long duration ballons. Some technical difficulties are described in the paper along with solutions to overcome them, so one can expect that this system will provide detailed measurements during the planned campaign, which will provide new insights into the structure of the TTL. Consequently, this paper is within the scope of AMT and I recommend publication.
The paper is well structured and fluently to read, some mostly minor comments are listed in the following.

**1)** Page 2 and 3, Figure 1: Please give et a better overview of how the entire system is composed, as I must confess that I don't get an idea of it from the current presentation. I recommend to indicate where the different components of the payload are located in more detail in Figure 1 (maybe add a photo ?) and the text (pages 2 and 3): RACHuTS – the primary balloon gondola, the smaller subgondola, profiler with sensors, reel system, upper gondola (the 'Euros'), lower gondola (the 'Zephyr').

**2)** Page 3, line 70: you might add here a recent study summarizing water vapor and ice cloud measurements from aircraft experiments, including experiments in the TTL:

Krämer, M., Rolf, C., Spelten, N., Afchine, A., Fahey, D., Jensen, E., Khaykin, S., Kuhn, T., Lawson, P., Lykov, A., Pan, L. L., Riese, M., Rollins, A., Stroh, F., Thornberry, T., Wolf, V., Woods, S., Spichtinger, P., Quaas, J., and Sourdeval, O.: A microphysics guide to cirrus – Part 2: Climatologies of clouds and humidity from observations, Atmos. Chem. Phys., 20, 12569–12608, https://doi.org/10.5194/acp-20-12569-2020, 2020.

**3)** Section 2, Instrument Description: It would be convenient for the reader if the panels and colors of the components shown in Figs. 2 and 3 would be also noted in the text.

**4)** Section 2.2.3, RACHuTS Optical Particle Counter:

How is the inlet design of the ROPC? The sampling efficiency of particles larger than ~ 3-5 µm is probably biased by the angle of the inlet to the airflow and the velocity differences between in- and outside of the inlet. I recommend to mention these effects.

The low upper size limit of 10 µm in diameter limits the ice cloud detection, as ice clouds often consist of larger ice particles (the total range is ~ 3 – 1000 µm in the TTL). This could be also noticed in the section.

**5)** Page 9, line 267-68: ' RACHuTS was hosted on the 'TTL3' configuration of the Zephyr, sharing the gondola with an LPC ...'

I'm a little puzzled about the location and characteristics of the LPC (see also comment 1) ) ?
A brief description (on page 5 or in section 2.2.3) would be helpful.

**6)** Page 10, Figure 5: Labeling of the panels ('a' and 'b') are missing.

**7)** Page 12, line 373: Figure 7 could be referenced already here. In the caption Figure 7, I would mention the water vapor contamination from outgassing from the balloon and gondola surfaces at < 67 hPa. Without this explanation the profile looks strange.

**8)** Page 13, Section 3.4 ROPC Performance: As noted earlier (comment 5), a brief description of the LPC (including sampling characteristics) would be helpful, in particular as it is now used for comparison with the ROPC. Could differing sampling characteristics be responsible for the difference between the instruments (see also comment 4) ? Adding a Figure about the instrument comparison would not unnecessarily lengthen the paper.

**9)** Figure 8: Labeling of the panels ('a' to 'd') are missing.

---

## Referee Comment (RC2) · Masatomo Fujiwara (Referee) · 29 Nov 2020

This paper describes the technical details of the Reeldown Aerosol Cloud Humidity and Temperature Sensor (RACHuTS) instrument system used with the CNES's long-duration balloons under the Stratéole 2 mission, and presents some results from an engineering flight along the equator for 101 days. The RACHuTS instrument system is a very interesting and challenging one, but at the same time is a very promising one. The paper is well written, and the example of scientific measurements is very interesting. I think that the paper is within the scope of the Atmospheric Measurement

Techniques and will be acceptable for publication after considering my minor comments listed below.

Lines 74-76: There were several water vapor sounding campaigns in the tropical Pacific and in the tropical Indian Ocean including Indonesian maritime continent; some of the papers are listed below.

Fujiwara, M., H. Vömel, F. Hasebe, M. Shiotani, S.-Y. Ogino, S. Iwasaki, N. Nishi, T. Shibata, K. Shimizu, E. Nishimoto, J. M. Valverde-Canossa, H. B. Selkirk, and S. J. Oltmans (2010), Seasonal to decadal variations of water vapor in the tropical lower stratosphere observed with balloon-borne cryogenic frostpoint hygrometers, Journal of Geophysical Research, 115, D18304, doi: 10.1029/2010JD014179.

Hasebe, F., Y. Inai, M. Shiotani, M. Fujiwara, H. Vömel, N. Nishi, S.-Y. Ogino, T. Shibata, S. Iwasaki, N. Komala, T. Peter, and S. J. Oltmans (2013), Cold trap dehydration in the Tropical Tropopause Layer characterized by SOWER chilled-mirror hygrometer network data in the Tropical Pacific, Atmospheric Chemistry and Physics, 13, 4393-4411, doi: 10.5194/acp-13-4393-2013.

Inai, Y., F. Hasebe, M. Fujiwara, M. Shiotani, N. Nishi, S.-Y. Ogino, H. Vömel, S. Iwasaki, and T. Shibata (2013), Dehydration in the tropical tropopause layer estimated from the water vapor match, Atmospheric Chemistry and Physics, 13, 8623-8642, doi: 10.5194/acp-13-8623-2013.

Suzuki, J., M. Fujiwara, T. Nishizawa, R. Shirooka, K. Yoneyama, M. Katsumata, I. Matsui, and N. Sugimoto (2013), The occurrence of cirrus clouds associated with eastward propagating equatorial $n\approx=\approx0$ inertio-gravity and Kelvin waves in November 2011 during the CINDY2011/DYNAMO campaign, Journal of Geophysical Research, 118(23), 12941-12947, doi: 10.1002/2013JD019960.

Lines 166-: Regarding the docking connector, is there potential icing issue? If the answer is no, that's OK; there is a recent paper by Jorge et al. (2020) whose discussions,

I thought, might have some relevance.

Jorge, T., Brunamonti, S., Poltera, Y., Wienhold, F. G., Luo, B. P., Oelsner, P., Hanumanthu, S., Sing, B. B., Körner, S., Dirksen, R., Naja, M., Fadnavis, S., and Peter, T.: Understanding cryogenic frost point hygrometer measurements after contamination by mixed-phase clouds, Atmos. Meas. Tech. Discuss., https://doi.org/10.5194/amt-2020-176, accepted, 2020.

Lines 177-: Does the Profiler have GPS position measurements? Or, the position information is taken by Zephyr? If the latter is true, please explain how to get the position information (in particular, the vertical position) at the measurement points. In Figures 5-8, in their captions, please specify which instrument is used to provide the vertical coordinate information.

Line 210: "to" should be "the"?

Lines 358-: In the near future, some close RS41 radiosonde sounding data at Singapore, at 1-second temporal resolution, may be available from the GRUAN data archive once the RS41 data product is certified (https://www.gruan.org/data/data-products). Also, again in the future, it would be interesting to make comparisons with GPS Radio Occultation temperature data.

Lines 382-: At the MLS 83 hPa level, we see that there is $\sim$0.5 ppmv difference on average between FLASH-B and MLS measurements, with MLS being lower. Are there any potential explanations for these differences? (Let me note that it seems that the results shown in the paper by Hurst et al. 2016 do not explain this.)

Hurst, D. F., Read, W. G., Vömel, H., Selkirk, H. B., Rosenlof, K. H., Davis, S. M., Hall, E. G., Jordan, A. F., and Oltmans, S. J.: Recent divergences in stratospheric water vapor measurements by frost point hygrometers and the Aura Microwave Limb Sounder, Atmos. Meas. Tech., 9, 4447–4457, https://doi.org/10.5194/amt-9-4447-2016, 2016.

Line 415: Is it really possible that cirrus cloud particles can exist above the cold-point

tropopause where the relative humidity is far less than 100% RHi (actually it is $\sim$50% RHi)?

Line 418: It may be actually surprising that the water vapor mixing ratio is low and constant at $\sim$3 ppmv between the cold-point tropopause and $\sim$1 km below. But, by looking at various profiles presented by Fujiwara et al. (2010), that might happen, although that may be rather rare.

Figure 1: Please explain (either in the caption or on the figure) which is Euros, which is Zephyr with the winch, and which is the Profiler with sensors. I happened to find the following presentation, and I found that the left-hand-side figure at Slide 4 was useful for me to understand the system: https://sites.google.com/umn.edu/2019-scientific-ballooning-tec/program ("Reeldown Instrument System Design for Atmospheric Profiling on Long-Duration Super Pressure Balloon Platforms" by St. Clair).

Figure 2: Could you add the explanation about how the line goes from (b)-(c) through (e) to (f)?

Figure 6: Is the vertical axis for radiosonde data geopotential height? Is the vertical axis for RACHuTS data also geopotential height (e.g., that has been converted from GPS geometric height)?

---

## Author Comment (AC2) · 26 Jan 2021

[revised manuscript text omitted]
for larger particles is not well characterized.   The ROPC was primarily included in the payload to detect super-micron cirrus cloud ice particles, while the measurement of the number density of ice particles needs further validation during flight, there is high confidence in the ability to detect cloud particle ($d > 1\mu m$) at concentrations above $0.1\ L^{-1}$ with limited quantitative confidence.

**3.     Stratéole 2 Engineering Flight**

[revised manuscript text omitted]

**Figure 8: RACHuTS profile measurement collected in the Eastern Pacific. All plots are a function of GPS altitude (left to right): (a) TSEN air temperature, note the clear cold point (185K) at 18.2km and wave driven temperature structures near 18.4 and 18.9km on ascent. (b) ROPC Aerosol/Cloud concentration, not the presence of large particles (d > 3.0μm), likely cirrus cloud ice particles, near the cold point (18.2km) and associated with negative temperature anomalies at 18.4 and 18.9km. (c) FLASH-B water vapor mixing ratio, note the relatively constant mixing ratio across the cold point (18.2km). (d) Calculated relative humidity over ice, near the cold point super saturations exceed 150% suggesting minimal dehydration through cloud formation.**

---

## Author Comment (AC1)

**Response to Reviewers for 'A Reel-Down Instrument System for Profile Measurements of Water Vapor, Temperature, Clouds and Aerosol Beneath Constant Altitude Scientific Balloons' by Lars Kalnajs et al.**

The authors would like to thank both reviewers for their time and useful input to this manuscript. We have addressed the reviewer's comments through minor modifications to the manuscript which we will detail point-by-point in this response to reviewers.

**Response to Review by Masatomo Fujiwara:**

We appreciate Dr Fujiwara's positive review of the paper and useful additions.   In response to the minor comments:

*Lines 74-76: There were several water vapor sounding campaigns in the tropical Pacific and in the tropical Indian Ocean including Indonesian maritime continent; some of the papers are listed below.*

Thank you for these references, it is important to acknowledge the prior measurements in this challenging region.   We have modified the manuscript to include some of these references (line 76)

*Lines 166-: Regarding the docking connector, is there potential icing issue? If the answer is no, that's OK; there is a recent paper by Jorge et al. (2020) whose discussions, I thought, might have some relevance.*

This is an astute observation and useful reference – we may have experienced an icing issue early in the flight that impacted the ability to dock the Profiler early in the campaign.   This was possibly due to ice accumulated on ascent through wet clouds in the troposphere that then sublimated in the lower stratosphere.   This was part of the motivation for including a Radio Frequency link to the Profiler in future campaigns.   We have added this possible issue to section 3.0 of the manuscript (line 281)

*Lines 177-: Does the Profiler have GPS position measurements? Or, the position information is taken by Zephyr? If the latter is true, please explain how to get the position information (in particular, the vertical position) at the measurement points. In Figures 5-8, in their captions, please specify which instrument is used to provide the vertical coordinate information.*

The Profiler does include a GPS receiver, as briefly mentioned on line 185.   We have updated the figure captions to indicate the vertical ordinate is GPS derived altitude (figures 5, 6, and 8).

*Line 210: "to" should be "the"?*

Fixed in the manuscript (line 213)

*Lines 358-: In the near future, some close RS41 radiosonde sounding data at Singapore, at 1-second temporal resolution, may be available from the GRUAN data archive once the RS41 data*

*product is certified (https://www.gruan.org/data/data-products). Also, again in the future, it would be interesting to make comparisons with GPS Radio Occultation temperature data.*

Thank you for this suggestion – we are always looking for comparative data points, both for the completed flight, and for the upcoming science flights.   We will look out for the high-resolution Singapore Sonde data in the GRUAN database and GPS RO data.

*Lines 382-: At the MLS 83 hPa level, we see that there is ~0.5 ppmv difference on average between FLASH-B and MLS measurements, with MLS being lower. Are there any potential explanations for these differences? (Let me note that it seems that the results shown in the paper by Hurst et al. 2016 do not explain this.)*
*Hurst, D. F., Read, W. G., Vömel, H., Selkirk, H. B., Rosenlof, K. H., Davis, S. M., Hall, E. G., Jordan, A. F., and Oltmans, S. J.: Recent divergences in stratospheric water vapor measurements by frost point hygrometers and the Aura Microwave Limb Sounder, Atmos. Meas. Tech., 9, 4447–4457, https://doi.org/10.5194/amt-9-4447-2016, 2016.*

Our intent with Figure 7 is demonstrate that the FLASH-B water vapor results are within the range of MLS water vapor measurements in a similar area and time frame (within 300km and within 12 hours) and therefore appear to be reasonable.   Our goal is not to suggest that there is sufficient data or co-location to provide a robust statistical comparison between the in situ and satellite measurements, and we have not averaged the FLASH-B data over the MLS averaging kernels.   We do not have an explanation for a possible bias between FLASH-B and MLS, and it is not clear that the bias is significant relative to instrument uncertainty. Figure 1 (in this Response to Reviewers) is a revised version of Figure 7 from the manuscript with the addition of the mean FLASH-B water vapor mixing ratio.  From this figure, MLS is reporting a mixing ration of 3.1 ppmv at 83hPa, and FLASH-B has a median value of 3.4 ppmv at the same pressure level.  Thus, the difference between the median FLASH-B water vapor mixing ratio at 83 hPa is within the combined uncertainty of MLS (9% at 83hPa) and FLASH-B (< 10%).

[Figure]

Figure 1 Comparison of the frequency of FLASH-B water vapor measurements from the 55 descent profiles with the average of closest Microwave Limb Sounder water vapor measurements within 12 hours of each profile (adapted from manuscript Figure 7). The black points show the median FLASH-B water vapor mixing ratio from all profiles at each pressure level.

*Line 415: Is it really possible that cirrus cloud particles can exist above the cold-point tropopause where the relative humidity is far less than 100% RHi (actually it is ~50% RHi)?*

This is one of the more interesting observations in the data set, and certainly will be subject to further analysis in subsequent publications. There are a few other observations of cirrus cloud particles above the tropopause and with RHi < 100% (down to 48% in some cases) from aircraft measurements in the lower most tropical stratosphere (Corti et al., 2008; de Reus et al., 2008). Given the prior observations, this was not cause for concern over the RACHuTS measurements, and thus we have left further discussions of these features to future water vapor focused works.

*Line 418: It may be actually surprising that the water vapor mixing ratio is low and constant at ~3 ppmv between the cold-point tropopause and ~1 km below. But, by looking at various*

*profiles presented by Fujiwara et al. (2010), that might happen, although that may be rather rare.*

Agreed.  This also has some interesting ramifications for 'cold trap' dehydration at the tropopause and warrants further investigation.  As with the ice particles above the tropopause, this will be discussed in detail in future water vapor focused works.

*Figure 1: Please explain (either in the caption or on the figure) which is Euros, which is Zephyr with the winch, and which is the Profiler with sensors. I happened to find the following presentation, and I found that the left-hand-side figure at Slide 4 was useful for me to understand the system: https://sites.google.com/umn.edu/2019-scientificballooning-tec/program ("Reeldown Instrument System Design for Atmospheric Profiling on Long-Duration Super Pressure Balloon Platforms" by St. Clair).*

We have modified Figure 1 to include a diagram similar to the figure from Alex St. Clair's presentation.

*Figure 2: Could you add the explanation about how the line goes from (b)-(c) through (e) to (f)?*

We have added text to the caption to explain the routing of the line through the pulley system.

*Figure 6: Is the vertical axis for radiosonde data geopotential height? Is the vertical axis for RACHuTS data also geopotential height (e.g., that has been converted from GPS geometric height)?*

The radiosonde altitude in Figure 6 has been converted from geopotential height to geometric height.   As these profiles are very near the equator, the difference between these altitudes is less than 50m.

**Response to Review by Anonymous Referee #2:**

We also appreciate the detailed comments and suggestions from the second reviewer.   Here we detail how these have been addressed:

*1) Page 2 and 3, Figure 1: Please give et a better overview of how the entire system is composed, as I must confess that I don't get an idea of it from the current presentation. I recommend to indicate where the different components of the payload are located in more detail in Figure 1 (maybe add a photo ?) and the text (pages 2 and 3): RACHuTS – the primary balloon gondola, the smaller subgondola, profiler with sensors, reel system, upper gondola (the 'Euros'), lower gondola (the 'Zephyr').*

We have added an overall flight train diagram to Figure 1 to clarify the configuration and names of the two main gondolas and the RACHuTS Profiler sub-gondola.

*2) Page 3, line 70: you might add here a recent study summarizing water vapor and ice cloud measurements from aircraft experiments, including experiments in the TTL: Krämer, M., Rolf, C., Spelten, N., Afchine, A., Fahey, D., Jensen, E., Khaykin, S., Kuhn, T., Lawson, P., Lykov, A., Pan, L. L., Riese, M., Rollins, A., Stroh, F., Thornberry, T., Wolf, V., Woods, S., Spichtinger, P., Quaas, J., and Sourdeval, O.: A microphysics guide to cirrus – Part 2: Climatologies of clouds and humidity from observations, Atmos. Chem. Phys., 20, 12569–12608, https://doi.org/10.5194/acp-20-12569-2020, 2020.*

Excellent suggestion, we have added the reference (line 71).

*3) Section 2, Instrument Description: It would be convenient for the reader if the panels and colors of the components shown in Figs. 2 and 3 would be also noted in the text.*

We have added the component label and color coding to the text.

*4) Section 2.2.3, RACHuTS Optical Particle Counter: How is the inlet design of the ROPC? The sampling efficiency of particles larger than ~ 3-5 μm ism is probably biased by the angle of the inlet to the airflow and the velocity differences between in- and outside of the inlet. I recommend to mention these effects. The low upper size limit of 10 μm ism in diameter limits the ice cloud detection, as ice clouds often consist of larger ice particles (the total range is ~ 3 – 1000 μm ism in the TTL). This could be also noticed in the section.*

The inlet tube for the ROPC is a small internal diameter (2mm) stainless steel tube, coupled to the OPC itself through a short section of conductive flexible tubing.   The inlet itself protrudes from the Profiler housing approximately 5cm and the air velocity through the inlet is approximately 8 m s$^{-1}$.   As the relative wind velocity and orientation of the inlet is unknown, a detailed analysis of the sampling efficiency of the inlet is not possible while the profiler is deployed.  However, while docked, the relative wind speed is essentially zero due to the Lagrangian nature of the balloon, and thus we expect comparisons between the ROPC and the separate LPC aerosol instrument to be relatively robust.   The 10 μm upper size limit is the largest size for which instrument can determine size, particles larger than this will still be detected, and included in the largest size bin.
Due to these limitations, we consider the cloud/aerosol data to be qualitative for large particles rather than quantitative.  Thus, the ROPC functions as a cirrus cloud detector, as opposed to providing quantitative data as to ice volume.   We have added text to section 2.2.3 to clarify this point, and have included the Profiler aerosol inlet location to Figure 1.

*5) Page 9, line 267-68: ' RACHuTS was hosted on the 'TTL3' configuration of the Zephyr, sharing the gondola with an LPC ...' I'm a little puzzled about the location and characteristics of the LPC (see also comment 1) ) ? A brief description (on page 5 or in section 2.2.3) would be helpful.*

 Figure 1 has been revised to add a diagram showing the location of the LASP Particle Counter (LPC) in the main gondola.   We have added text to section 3.4 to provide more detail on LPC in the context of the ROPC to LPC comparisons.

*6) Page 10, Figure 5: Labeling of the panels ('a' and 'b') are missing.*

Fixed in the manuscript.

*7) Page 12, line 373: Figure 7 could be referenced already here. In the caption Figure 7, I would mention the water vapor contamination from outgassing from the balloon and gondola surfaces at < 67 hPa. Without this explanation the profile looks strange.*

We have added the reference to figure 7 to the section discussing water vapor contamination as well as identifying the contaminated region in the caption of Figure 7.

*8) Page 13, Section 3.4 ROPC Performance: As noted earlier (comment 5), a brief description of the LPC (including sampling characteristics) would be helpful, in particular as it is now used for comparison with the ROPC. Could differing sampling characteristics be responsible for the difference between the instruments (see also comment 4) ? Adding a Figure about the instrument comparison would not unnecessarily lengthen the paper.*

We have added to the description of LPC in Section 3.4.  The difference in sampling characteristics (flow rate, inlet velocity, residence time) may be responsible for some of the differences between the instruments, however it should be noted that these comparisons occurred at zero relative wind speed and only for sub-micron particles, which should be less impacted by inlet characteristics.

*9) Figure 8: Labeling of the panels ('a' to 'd') are missing.*

Fixed in the manuscript.

References Cited:

Corti, T., Luo, B. P., Reus, M. de, Brunner, D., Cairo, F., Mahoney, M. J., Martucci, G., Matthey, R., Mitev, V., Santos, F. H. dos, Schiller, C., Shur, G., Sitnikov, N. M., Spelten, N., Vössing, H. J., Borrmann, S. and Peter, T.: Unprecedented evidence for deep convection hydrating the tropical stratosphere, Geophys. Res. Lett., 35(10), https://doi.org/10.1029/2008GL033641, 2008.

de Reus, M., S, B., Heymsfield, A., R, W., C, S., Mitev, V., Frey, W., Kunkel, D., Kürten, A., Curtius, J., Sitnikov, N., A, U. and Ravegnani, F.: Evidence for ice particles in the tropical stratosphere from in-situ measurements, Atmospheric Chem. Phys. Discuss., 9, https://doi.org/10.5194/acp-9-6775-2009, 2008.